# Protocol for a qualitative study exploring the lived experience of hearing loss and patient reported experience in the UK: the HeLP study

Helen Pryce,[1] Sian Karen Smith [ID],[1] Georgina Burns-O'Connell,[1] Rachel Shaw [ID],[2,3] Saira Hussain,[1] Jonathan Banks [ID],[4,5] Amanda Hall [ID],[1,6] Rebecca Knibb [ID],[2] Rosemary Greenwood,[7] Jean Straus[8]

For numbered affiliations see end of article.

**Correspondence to**
Dr Helen Pryce;
h.pryce-cazalet@aston.ac.uk

## ABSTRACT

**Introduction** Worldwide, hearing loss is a significant public health issue and one of the most common chronic health conditions experienced by older adults. Hearing loss is associated with communication difficulties, social withdrawal, isolation and lower quality of life. Although hearing aid technology has improved significantly, the workload of managing hearing aids has increased. The aim of this qualitative study is to develop a novel theory of people's lived experience of hearing loss across the lifespan.

**Methods** Eligible participants will be young people and adults aged 16 years and above who have a hearing loss and carers/family members of people with a hearing loss. This study will use individual, in-depth face-to-face or online interviews. With participants' permission, interviews will be audio-recorded and transcribed verbatim. A grounded theory approach to concurrent data gathering and analysis will develop grouped codes and categories and link these to provide a novel theory to describe the experience of hearing loss.

**Ethics and dissemination** The study was approved by the West of Scotland Research Ethics Service (approval date: 6 May 2022 ref: 22/WS/0057) and the Health Research Authority and Health and Care Research Wales Approval (approval date: 14 June 2022; IRAS project ID: 308816). The research will inform the development of a Patient Reported Experience Measure to improve the information and support given to patients. Findings will be disseminated through peer-reviewed articles and at academic conferences, as well as to our patient and public involvement groups, healthcare professionals, audiology services and local commissioners.

## STRENGTHS AND LIMITATIONS OF THIS STUDY

⇒ Our qualitative interview study, informed by grounded theory approaches, will generate a novel theory on the lived experience of hearing loss across the life course.
⇒ Different patient, public involvement groups will be involved throughout the project to guide the logistics of conducting the research and support data analysis procedures.
⇒ Purposive sampling via different clinical sites and non-clinical routes (eg, lip reading classes, social media) will help to ensure diverse experiences and views are captured.
⇒ The study will be restricted to people living in the UK which may limit the applicability of the findings to other settings.
⇒ This qualitative study part of a wider study that will inform the development of the first Patient Reported Experience Measure internationally to enable audiologists to identify the individualised and effective support.

## INTRODUCTION

Hearing loss is a chronic health condition that affects an estimated 430 million adults worldwide.[1] With the ageing population, hearing loss is set to affect 700 million (one in ten) of the global population by 2050. In the UK, hearing loss affects around 12 million (one in five) adults, which is predicted to increase to 14.2 million by 2035.[2] The prevalence of hearing loss rises sharply with age. In the UK, around 40% of adults aged over 50 years old live with hearing loss, increasing to 70% of older adults over the age of 70.[3] Hearing loss perpetuates socioeconomic inequality with its greatest prevalence associated with lower socioeconomic status.[4,5]

The implications of hearing loss are well-established and far reaching. People with hearing loss may feel ashamed or frustrated about their difficulty communicating and limit their social interactions.[6] This separation can lead to social isolation, and feelings of loneliness, anxiety, depression, as well as fewer opportunities for engaging in listening and communicating.[7–10] Hearing loss is linked with other chronic health conditions, including arthritis, cancer, cardiovascular risk factors, diabetes, stroke, visual impairment

and mobility issues.[11] There is also growing evidence that hearing loss increases the risk of developing dementia, although the mechanisms linking the two are as yet unclear.[12–14]

There is no permanent cure for hearing loss, and the main form of clinical intervention is wearing hearing aids. Using hearing aids has been shown to improve hearing function for mild to moderate hearing loss, as well as positively influence quality of life through enhancing communication.[15–17] Despite these benefits, hearing aid use is low and approximately 30% of people report wearing hearing aids some of the time and 20% do not wear them at all.[18] Furthermore, many people live with hearing loss symptoms for a long time (on average, up to 10 years) before seeking help.[19] Understanding the reasons why hearing aid usage is low and gaining insight into why people do not seek help are important research priorities.

The illness burden of hearing loss is considerable. Hearing loss makes it hard work to communicate and increases risks in all aspects of life.[6 20] Adjusting to hearing aids can be particularly tiring and frustrating as it takes time to adapt to amplified sounds and to learn how amplified sounds correspond to previous auditory memories.[21] All the treatment work is devolved to the patient, including wearing the device each day, and replacing the fine tubing monthly, the batteries weekly, keeping the aid at the right volume settings. The stigma about visible deficits combined with the association of age with wearing hearing aids often means that people do not want hearing aids.[22–25] Although hearing aid technology and assistive devices have advanced, the workload of managing hearing aids has increased significantly (eg, learning how to use Bluetooth to pair devices, sequencing turning on and off different devices).[26] Hearing aid non-use (including people who do not have hearing aids and those who have hearing aids but find them difficult to manage) has been attributed as a problem of the hearing aid user, their motivation or ability, rather than a reflection of an onerous workload. The trade-off between the benefits of wearing hearing aids versus the work required to use them is likely to vary across the life course.

The 'burden of treatment' theory is an increasingly useful way to consider the relative benefits of allocating health resources.[27] Health services systematically transfer work to patients to manage long-term conditions.[28 29] Previous research has focused on giving patients more to do, for example, education and skills training[30 31] which further enhances inequality as not all adults are equally able to undertake the work required.[5] Work with younger people with hearing loss transitioning into adult audiology services emphasises the burden in navigating audiology services.[32] Similarly, in older age, the benefits of hearing aids may be outweighed by the burden of work required to access and use them.[22] This work is out of view of audiology services and thus overlooked when planning how to provide care.

In the UK, audiology services are commissioned differently in different parts of the country, with the introduction of 'Any Qualified Provider' (AQP) services meaning that large corporations and optical chains have been awarded NHS contracts to provide audiology services in England.[33] AQP service providers are primarily funded to provide hearing aids, so there is little time (if any) to identify patients' values and to engage in shared decision making about preferred treatment options.[34 35] At the same time, guidelines from the National Institute of Clinical Excellence propose that patients be fitted with hearing aids and then given motivational interviewing to increase their motivation to use them.[36] More experiential evidence is needed to understand the reasons why people do and do not use hearing aids and the factors that encourage their use.

It is critical that service development and delivery are grounded in experiential evidence to develop a good understanding of the daily illness and treatment burdens of hearing. Previous work shows that while there are benefits in aspects of communication from hearing aids, there are specific barriers to use that derive from social identity preservation and stigma of hearing loss (and hearing aids as a visible marker of hearing loss).[6 20 22 35] Our own work suggests that people seek help with hearing to gain information and support rather than hearing aids per se.[22] However, changes in commissioning of audiology services have resulted in patients being prescribed hearing aids rather than engaging in shared decision making.[35]

Further research is required to understand how people make the trade-offs at different life stages between living with a hearing loss and using a hearing aid to manage the hearing loss. Such research will enable audiology services to be sensitive to patient experience and promote shared decision making. Therefore, the overall goal of the current study is to provide a deeper understanding of the lived experiences of coping with hearing loss (with and without hearing aids) at different stages of life, from the perspectives of patients and their families. This study forms part of a larger study—the Hearing Loss and Patient Reported Experience study (HeLP study)—the first study of its kind to prioritise patients' lived experience of hearing loss to inform decision-making in audiology care planning and services. The findings from this qualitative work will inform the development of potential items for a novel Patient Reported Experience Measure to be used in clinical practice to understand the work involved in living with hearing loss ('illness work') and hearing aids ('treatment work').

### Aims and objectives

Drawing on grounded theory methodology, the aim of this study is to develop a comprehensive novel theory that has the potential to explain variation in people's lived experiences of hearing loss. The objectives are to:

1. Develop a theoretical understanding of how coping with hearing loss ('illness work') and accessing and living with hearing aids ('treatment work') is experienced and negotiated alongside personal and sociocultural contexts.

2. Understand how individuals adapt to and manage hearing aids.
3. Explore the reasons why hearing aid non-use occurs.
4. Explore patients' experiences of accessing audiology services and support.

## METHODS AND ANALYSIS
### Study design
This qualitative interview study will be informed by Strauss and Corbin's pragmatic approach to grounded theory.[37 38] This allows for (and recognises the likelihood of) researchers to bringing prior knowledge to the process of refining theoretical concepts. Grounded theory methodologies are particularly well suited to developing an understanding of patterns across contrasting cases and processes that influence those patterns. In comparison to other qualitative methodologies, grounded theories offer explanatory potential as well as description.[39] In this work, we want to identify the work involved in hearing loss and hearing aid use, how it is experienced, and the factors informing variation in this experience. This study will be conducted and reported in accordance with the Consolidated criteria for Reporting Qualitative research checklist,[40] and the Guideline for Reporting and Evaluating Grounded Theory Research Studies.[41]

### Research team and reflexivity
For reflexivity, the study team comprises two female clinician-researchers (HP: an academic and hearing therapist with clinical experience in audiological rehabilitation and grounded theory methodology; SH: a clinical scientist, teaching fellow and researcher in audiology), three female researchers (GB-OC: a teaching fellow and researcher with a background in sociology and interest in hearing research; RS: a health psychologist with experience of applied qualitative research; SKS: a health psychologist with personal interest in hearing research) and a female patient and public involvement (PPI) lead (JS) who is an expert in audiology by experience and writer. All the team either have first-hand experience of hearing loss or are closely related to someone who does.

Participants will be interviewed by either HP, GB-OC, SH or SKS. At the start of each interview, the interviewers will undertake a 'working alliance' discussion agreeing participant and researcher roles, timing and nature of the interview. This will include the researcher's interest and motivations in the topic.

The study team will meet fortnightly to reflect on their role in shaping data collection and compare and contrast their interpretations of the data. Throughout the study, the wider HeLP research group and core PPI members will discuss interpretations of the findings for triangulation and work together to develop and link theoretical categories for the explanatory model.

### Study Steering Committee group
The Study Steering Committee will have overall oversight of the study and provide advice to the study research team,

---

> **Box 1   Eligibility criteria for study participation and sample**
>
> **Inclusion criteria**
> ⇒ Young adults (aged 16–29 years): transitioning from paediatric to adult services and negotiating independence. We will aim to recruit 4–6 young people, including those with additional disabilities such as learning disabilities, and up to four parents.
> ⇒ Adults aged 30–49 years: managing hearing loss while pursuing career/work and family life; 4–6 people with diagnosed hearing loss either using or not using hearing aids, plus 2–4 partners/relatives.
> ⇒ Adults aged 50–79 years: noticing symptoms of hearing loss for the first time; 4–6 hearing aid users and 5–8 non-hearing aid users.
> ⇒ Adults aged 80 years to end of life: most likely to have hearing loss. Other health conditions likely and changes in living situation possible; 4–6 hearing aid users, 4–6 non-hearing aid users and 2–4 carers from residential care settings.
>
> **Exclusion criteria**
> ⇒ Volunteers without close connection to or experience of hearing loss.

funder and sponsor. Our steering group will comprise eight independent members, including: one international collaborative representative, two experts by experience, three clinicians working in audiology in different parts of the country, one social science researcher and chaired by a health commissioner with a particular interest in person centred care. The steering group will meet every 6 months and oversee delivery milestones and reporting.

### Study setting
The study is being conducted in the UK with participants (adults with hearing loss and carers, family members or parents) recruited from three clinical sites in England (Bristol and Bath) and Scotland (Tayside) and non-clinical groups (eg, lip reading classes, social care settings).

### Participants and recruitment
Participants are young people and adults aged 16 years and over and individuals who have experience of hearing loss directly or as a carer, partner or parent. Eligibility criteria (box 1) are intentionally broad to recruit participants of different ages across the life course as it is acknowledged that hearing loss experience and needs will vary at different life stages. The geographical spread of the different clinical sites (rural, urban and semi-urban) will enable us to achieve socioeconomic variation and find cases across the age range with contrasting features in terms of sex, income, housing and clinical needs.

Drawing on the principles of Strass and Corbin's grounded theory approach, we will be using purposive sampling to provide maximum variation of cases to generate rich data from different perspectives which will allow us to target under-represented groups. We will examine the experience of different age groups and identify similarities and differences between and across the life stages This will enable us to build up an understanding of

living with hearing loss (with and without hearing aids) across the life course.

Potentially eligible participants will be made aware of the study through advertisements. Clinical sites will advertise the study to new and existing patients via word-of-mouth, invitation flyers or posters displayed in waiting/reception areas. Participants will also be recruited through non-clinical routes including care homes, supported living centres and community groups. Advertisements will also be distributed via leads of tinnitus groups, teachers of lip-reading classes and staff in residential care settings. Details of the study will be circulated on social media (eg, Twitter) and the study link hosted on Aston University website with the aim of recruiting participants with diverse experiences and backgrounds. In addition, our PPI collaborator in South Asian community groups and care homes will prompt snowball sampling in which word of mouth advertisement will encourage potential participants to contact the study team. In accordance with our PPI group in the South Asian community regarding recruitment of participants who are not help seekers, we will recruit participants from local mosques and women's groups in the local area (eg, exercise groups for older Asian women and mindfulness group for Asian women). Previous research experience informed our sample estimate of approximately 25–38 patients and 8–12 family members/carers (see box 1) to capture a range of experiences and achieve sufficient information power, but this will be reviewed as data collection progresses.

Interested participants will directly contact the HeLP study team via email, phone or post. The researchers will then provide potential participants with the participant information sheet (online supplemental file). Informed consent will be obtained from researchers trained in Good Clinical Practice and experienced with research. Recruitment commenced June 2022 continuing through April 2023. At the time of writing, 18 participants have been recruited and data collection is ongoing.

### Data collection

Data will be collected using qualitative, semi-structured interviews by the study team (GB-OC, HP, SKS, SH) who are all experienced in qualitative research. Participants will have the option for the interview to take place at their preferred location including online (via Microsoft Teams), or at their home, the university or audiology department in a quiet and private space. An interview schedule been developed to explore the lived experience of managing hearing loss and hearing service use (box 2). It has been informed by a literature review, the expertise of the authors and input from the PPI groups. It covers: motivations to take part in the study, lived experience of hearing loss, challenges of hearing loss (emotions, practical efforts), the journeys and processes of referral and diagnosis, experiences of audiology services and support, decision-making around whether to wear hearing aids and trade-offs made between managing with and without hearing aids. Open-ended questions will be used to

---

### Box 2 Interview questions for patients and carers/family members

**Questions for people with hearing loss**
⇒ Tell me a little bit about yourself?
⇒ Tell me your story with your hearing and why you were interested in taking part?
⇒ Can you tell me your thoughts and feelings about hearing loss?
⇒ What have you found difficult about having a hearing loss?
⇒ What have you done to manage your hearing loss?
⇒ Decision making around hearing aid use or non-use/uptake—tell me how you came to be using/not using your assistive listening devices/hearing aids explore reasons why/why not?
⇒ What is important to you when deciding whether to use hearing devices/aids?
⇒ Who else is important in helping you decide?
⇒ Has this changed over time?
⇒ What has been helpful to you?
⇒ Have you sought help from audiology services?
⇒ Tell me about your experience with using audiology services
⇒ Based on your experience—what do you think they ought to know/do that they currently do not do?

**Questions for family/carers**
⇒ Tell me what is it like for you managing hearing loss?
⇒ What is difficult for you about living with someone with hearing loss?
⇒ What if anything do you do differently as a result of hearing loss?
⇒ Have you had any involvement with audiology services?
⇒ What was that like?
⇒ What is challenging about using hearing aids (if your family member/cared for person has them)?

---

encourage the participant to talk in-depth and flexibly about their experiences (box 2). The expected duration of the interviews is around 1 hour. With participants' permission, the interviews will be recorded on a digital voice recorder or online (video and audio-recorded) using Microsoft Teams and supplemented with field notes to capture important information not captured by the audio-recording. We will loan specialist headsets to enable people with hearing loss to hear better on the phone and via online platforms. All transcripts will be anonymised, and pseudonyms will be used. Transcripts will not be returned to participants for comment and/or correction as we want to capture experiences and views at that particular moment in time.

### Data analysis

In line with Strauss and Corbin's grounded theory procedures, data analysis and interviewing will be iterative, in which early analysis and findings will be compared with inform ongoing interviews. We will keep field notes of reflections and thoughts following interviews to support this. While the notion of data 'saturation' has been widely critiqued as a neo-positivist indication of a singular 'truth'; there is recognition of the importance of capturing a range of views, based on different contexts.[42] The research team will review the data until we are confident that a wide range of experiences are captured and there is sufficient conceptual depth to generate meaningful

insights to address the research aims.[42] Meaning statements will be examined and every line of transcript will be coded by three coders (GB-OC, SH, SKS) to summarise key content (open coding). Codes will be compared across and between transcripts and will be collated into broader categories with properties and dimensions that capture the range of meanings (axial coding). Finally, the relationships between these categories will be explored to develop a conceptual model that explains the variation in experience (selective coding). Triangulation is a core part of our activity with multiple researcher and PPI members contributing a view to the interpretation which is justified by each researcher. We will consider core categories that offer explanatory potential to the paradigm and explain variation within the data set.[39] Our analysis will describe the range of 'illness work' of hearing loss and the 'treatment work' of using hearing aids and audiology services. We will also capture the variations that exist and the important influences that contrast different social, cultural and age groups.

## Rigour in analysis

We will ensure credibility to the research process through transparent decision making, including incorporating PPI views. The lead investigator (HP) and coinvestigator (RS) will supervise the analysis closely and ensure consistency. They will blind code a subset of transcripts and compare coding. The HeLP qualitative research team (GB-OC, HP, SH, JS, SKS) will develop and refine the conceptual model while interviews and analysis progress to understand the effort required to manage hearing loss and highlight the variations in patient need at each life stage. We will examine the fit of this model to previous models and existing theories, and we will encourage reflection from all researchers and PPI leads through discussion.

## Patient and public involvement

A major component of the HeLP study is the involvement of PPI groups. Specifically, two PPI leads (a researcher with PPI responsibility and a public member) will manage and coordinate the PPI activities, working closing with the study team. Our PPI leads have experience in communicating with people with complex communication needs. PPI activities include reviewing patient information sheets and consent forms, advising on recruitment and interview questions, checking analysis procedures and addressing any uncertainties in data interpretation We will consult our PPI groups to check that our process and early findings have not omitted any aspects of the experience (eg, labelling codes and categories, development of conceptual framework). Fieldnote diaries will record PPI engagement and outcomes.

Certain groups are more likely to be affected by hearing loss including people from South Asia (Indian, Pakistani and Bangladeshi communities), older adults in residential care and adults with learning disabilities. Our PPI strategy reflects the diverse and heterogeneous nature of the population affected by hearing loss in the UK. We have directly targeted the following groups to expand our reach of PPI inclusion:

1. South Asian community groups: we have interest and agreement from these groups. We have also spoken to local religious centres, including but not limited to specific local Imams to enhance access to those from Muslim backgrounds. South Asian heritage people have both a disproportionately high risk of hearing loss and are under-represented in patient populations in the UK.[43 44]
2. Residential care homes: our PPI leads will meet with individuals in these settings (with whom we have existing relationships) face-to-face.
3. Aston PPI group: people in the local community who have experience with hearing loss, a group including student members who are younger.[6 18–39] They meet virtually and will advise via whatever form of communication suits them best (eg, phone, email, online messaging).
4. Bath PPI group: a well-established group of older adults who support and advise on service delivery and research. They meet face-to-face or via email.

Our initial PPI work informed the development of the proposal in a number of ways. First, our Aston group highlighted the need for researchers from the same culture and religious as participants to undertake PPI and research. As such, we will include a researcher with responsibility for PPI in marginalised groups and have recruited a researcher dedicated to South Asian groups. Second, our older group identified problems with hearing aids that inform the notion of 'work' (eg, struggles to fit hearing aids, learning skills such as battery changing). These issues will be explored further by targeting PPI and recruitment in residential care settings where access to audiology services can be challenging. Third, our older group noted the need to be flexible in where and how data gathering occurs and be mindful of travel and distance barriers for participants. This informed our plans to gather data in people's homes, at clinics, university and online/telephone. Finally, our younger group advised we purchase amplifier headsets which could be sent out to participants for online/phone use, if required.

## ETHICS AND DISSEMINATION
### Ethical approval

This study was reviewed and approved by the West of Scotland Research Ethics Service (approval date: 6 May 2022 ref: 22/WS/0057) and the Health Research Authority and Health and Care Research Wales (HCRW) Approval (approval date: 14 June 2022; IRAS project ID: 308816).

### Participant withdrawal

Each participant has the right to withdraw at any time during the study and information relating to all withdrawals will be recorded. If a participant wishes to discontinue, data collected up until that point will be kept and included in the analysis.

## Confidentiality

All data obtained from participants will be kept confidential. Participants will be anonymised with all identifying information removed. Participants' names will be replaced with pseudonyms.

## Benefits to participation

Participants may reflect on their experience of hearing loss (as a person living with hearing loss or as a carer/family member) and gain new understandings which might be empowering. Their insights will help inform how to better design audiology services and improve clinical practice and care.

## Assessment and management of risks

There is a possibility that discussing some topics will create distress. If at any point a participant feels distressed during the interview, they will be given the option to pause the recording and provide time to discuss the issue. The researcher will signpost participants to other services including patient liaison services within NHS providers. In the case of home interviews, researchers will follow the sponsor's (University Hospitals Bristol and Weston NHS Trust) lone working policy including plans to check in with another member of the research team before and after interviews.

## Storage of data

All study data will be stored on the university password protected server. Informed consent sheets (hard copies) containing personal information will be kept in a secure filing cabinet at the university only accessible to study staff and authorised personnel. Audio recordings of interviews will be conducted with encrypted digital recorders or MS Teams recording software. Once transcribed verbatim and checked for accuracy, recordings will be deleted. Data will be collected and retained in accordance with the Data Protection Act 2018 and General Data Protection Regulation standards. Once data analysis is completed, the study documents (paper and electronic) will be retained in a secure location for 5 years at Aston University. Personal data (eg, contact details) are held on a separate Aston University Box file and are separate from research data. Besides clinicians in Participant Identification Centres sending the advertisements to home addresses we will not access personal data.

## Informed consent

We will obtain written informed consent from all interview participants at the start of an arranged interview time. Participants will have had at least a week to consider the participant information sheet and ask questions in advance. Participants who volunteer for online meetings will post their consent form back in a prepaid envelope. The researcher will record the consent process verbally at the start of the interview as well.

Participation from adults with additional cognitive and communication difficulties are welcomed but will require capacity to consent for participation. In care home settings, where potential participants may experience dementia, researchers will check participants' capacity with care staff and only consent those who are able to provide consent independently. For this, the researcher will check that participants can understand the nature of the research and be able to retain the information to make a free choice about participation, and be capable of making this decision at a time it needs to be made.

## Dissemination of findings

We will prepare and publish articles in appropriate peer-reviewed journals and professional magazines (eg, British Academy of Audiology Audacity) to ensure the findings are shared with audiology healthcare professionals and academics. Findings will be disseminated via poster and oral presentations at relevant academic and non-scientific conferences, and social media. They will also be disseminated to PPI groups, audiology services and local commissioners.

**Author affiliations**
[1]Department of Audiology, Aston University College of Health and Life Sciences, Birmingham, UK
[2]Aston University School of Psychology, Birmingham, UK
[3]Aston Institute of Health & Neurodevelopment, Aston University College of Health and Life Sciences, Birmingham, UK
[4]University Hospitals Bristol and Weston NHS Foundation Trust, National Institute for Health Research, Applied Research Collaboration West, Bristol, UK
[5]Population Health Sciences, University of Bristol Medical School, Bristol, UK
[6]Children's Hearing Centre, University Hospitals Bristol NHS Foundation Trust, Bristol, UK
[7]Psychology Department, University Hospitals Bristol and Weston NHS Foundation Trust, Bristol, UK
[8]Patient and Public Involvement Representative and Lead, HeLP study, London, UK

**Acknowledgements** The sponsor for this study is the University Hospitals Bristol and Weston NHS Foundation Trust. Aston University (Applied Audiology research group, College of Health and Life Sciences) is the supporting institution.

**Contributors** HP is the lead researcher leading the protocol development, ethical approval, data collection, data analysis and dissemination. SKS wrote the first draft of the protocol, the final manuscript and prepared the submission. GB-OC, RS, SH, JB, AH, RK, RG and JS have provided feedback and support in the development of the protocol and study documents including the ethics application and interview schedule. All authors read, contributed to, edited and agreed the final manuscript.

**Funding** This study is supported by a National Institute for Health Research (NIHR) Health Services and Delivery Research programme (Funding stream REF NIHR 131597). JB is partly funded by National Institute for Health and Care Research Applied Research Collaboration West (NIHR ARC West) and NIHR Health and Social Care Delivery HS&DR (REF NIHR 131597).

**Competing interests** None declared.

**Patient and public involvement** Patients and/or the public were involved in the design, or conduct, or reporting, or dissemination plans of this research. Refer to the Methods section for further details.

**Patient consent for publication** Not applicable.

**Provenance and peer review** Not commissioned; externally peer reviewed.

of the translations (including but not limited to local regulations, clinical guidelines, terminology, drug names and drug dosages), and is not responsible for any error and/or omissions arising from translation and adaptation or otherwise.

**ORCID iDs**
Sian Karen Smith http://orcid.org/0000-0002-9541-2221
Rachel Shaw http://orcid.org/0000-0002-0438-7666
Jonathan Banks http://orcid.org/0000-0002-3889-6098
Amanda Hall http://orcid.org/0000-0001-8520-6005
Rebecca Knibb http://orcid.org/0000-0001-5561-0904

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
