## [Reviewer comments · BMJ Open]

ARTICLE DETAILS

TITLE (PROVISIONAL)	Protocol for a qualitative study exploring the lived experience of Hearing Loss and Patient Reported Experience in the UK: The HeLP study
AUTHORS	Pryce, Helen; Smith, Sian; Burns-O'Connell, Georgina; Shaw, Rachel; Hussain, Saira; Banks, Jonathan; Hall, Amanda; Knibb, Rebecca; Greenwood, Rosemary; Straus, Jean

VERSION 1 – REVIEW

REVIEWER	Henshaw, Helen University of Nottingham, School of Medicine
REVIEW RETURNED	05-Feb-2023

GENERAL COMMENTS	This manuscript reports the study protocol for a qualitative research study that is part of a larger programme of research. The team is highly experienced and well-placed to deliver this research. However, the purpose of the study, proposed methods, and outputs are somewhat unclear according to guidance for BMJ Open protocols. Abstract The abstract and strengths/limitations describe theory development generated from the lived experiences of a sample, and data collection/analysis occurring concurrently in line with grounded theory. However the study methods are not explicitly described in line with grounded theory (is this a grounded theory study? In which school of GT are the methods situated?) Introduction References 17 and 18 provide data from over a decade ago. The authors may wish to look at more recent estimates. For example: https://doi.org/10.1080/14992027.2020.1773550 Aims and objectives The text here differs from abstract, which suggests theory development. Here, the research aim is to develop a comprehensive contextualised explanation... Clarification required. Methods Informed by grounded theory, but not grounded theory? Drawing on the principles of grounded theory. It is not clear which aspects of grounded theory methodology the authors will use and which they will omit. I would strongly suggest outlining their plans for the use of appropriate tools/guidance within this protocol to inform the reporting of ensuing study results, such as GUREGT for theoretical aspects, and e.g. COREQ as good practice for the reporting of qualitative research.
---

	Study research team – the inclusion of names is non-standard. What would be more useful within the protocol would be to outline the methods by which the influence of the study team will be monitored, reflected upon, and considered when undertaking the research and agreeing study results. I believe there are a number of omissions in the methods that the authors have likely considered and planned for. I use the COREQ checklist as reference to suggest that the authors should pre-specify within this protocol what plans they have in place for:  1. Reflexivity - when/how/impact (see comments on methods, above) 2. Methodological orientation and theory (see comments on methods, above) 3. How will you define and report participant non-participation? 4. Plans for data management 5. Plans for data analysis including software, methods, raters, validation ... I am also unclear as to what the final output of the planned research will be (a developed theory or a contextual explanation - see comments on aims and objectives, above)? Further detail is required here. It is clear that this is part of a wider research programme. As such, describing the contribution of this research in relation to the wider research aims may be useful. PPI How has PPI shaped this study design/this protocol? Twitter Not sure these details are required? Appendices It is not a requirement to include the ethical approval letter as part of the published the study protocol. Although i have no doubt that the team have thought about all of the issues raised here in some detail, there is currently insufficient detail within this manuscript draft to convey that. The aim of the published protocol should be to provide a transparent, comprehensive and replicable outline of the study to be conducted. The final study results can then be reported in accordance with the protocol, and any deviations to protocol described and justified. The additional information requested will help to address that aim.
--	---

REVIEWER	Laird, Emma C. The University of Melbourne
REVIEW RETURNED	17-Feb-2023

GENERAL COMMENTS	Thank you for providing the protocol for your study exploring the lived experiences of people with hearing loss across a variety of sample groups. The use of the burden of treatment theory to frame "illness work" and "treatment work" seems highly applicable to
--

	hearing loss and the issues associated with the uptake of hearing devices. Qualitative studies in this area are few, and this study will add important additional information. I only have some minor comments to assist with comprehension of the protocol for those not familiar with qualitative study methods.  - Can you please provide more detailed information about how your transcripts will be coded? I understand a subset of transcripts will be coded and compared by two researchers but how many people are initially coding the interviews? - In the section Rigour in Analysis (pg 9-10) it reads like the theoretical connections will be formulated after the coding has been completed. Can you please clarify if this is what you mean or will you be developing the theory as the interview analysis progresses? -Please include information about theoretical saturation / data saturation, how do you know you have interviewed sufficient numbers? -Abstract: please specify acronym Patient and public involvement (PPI) when first used in abstract. - It wasn't clear why south Asian subgroups were singled out for recruitment, can you please provide rationale? Are you targeting only those who can speak English or will you be conducting interviews with interpreters? -It is recommended that the semi-structured interview guide be provided as supplementary material. I look forward to reading the outcomes of your research.
--	--

VERSION 1 – AUTHOR RESPONSE

Reviewer 1	
Abstract: The abstract and strengths/limitations describe theory development generated from the lived experiences of a sample, and data collection/analysis occurring concurrently in line with grounded theory. However the study methods are not explicitly described in line with grounded theory (is this a grounded theory study? In which school of GT are the methods situated?)	Thank you for raising this important point about the study's methods and methodology. Grounded theories are likely to offer insights, enhanced understanding and provide a meaningful guide to action. In this case we present a qualitative interview study informed by a grounded theory approach (e.g. an approach that extends analysis beyond thematic patterns description to a conceptual description and theory development. We are following an approach described by Strauss and Corbin (1990) which differs from the original Glaser and Straus (1967) in that it allows for (and recognises the likelihood of) researchers to bring

previous knowledge and theory development to the process of refining theoretical concepts. It does not assume that only the data set gathered during the research process matters and recognises the importance of prior expectation, knowledge etc.

Our research methods follow grounded theory procedures. We propose purposeful sampling, a series of coding procedures that extend constant comparative description to axial coding (linking concepts together) to provide explanation in the form of novel theory – not simply description. Unlike other constant comparative approaches we are looking at the data to refine a theoretical understanding that has the potential to explain variations between cases and therefore a grounded theory methodology is best suited to this task.

We now make this more explicit in the text:

Strengths and limitations of this study

Our qualitative interview study, informed by grounded theory approaches, will generate a novel theory on the lived experience of hearing loss across the life course.

Abstract (p.2)

The aim of this qualitative study is to develop a theoretical understanding of the lived experience of hearing loss (both with and without hearing aids) across the lifespan, and to navigate audiology services.

A grounded theory approach to concurrent data gathering and analysis will develop grouped codes and categories and link these to provide a novel theory to describe the experience of hearing loss.

Introduction (p.6)

Aims and objectives

	Drawing on grounded theory methodology, the aim of this study is to develop a comprehensive novel theory that has the potential to explain variation in people's lived experiences of hearing loss. The objectives are to:  1. Develop a theoretical understanding of how coping with hearing loss ('illness work') and accessing and living with hearing aids ('treatment work') is experienced and negotiated alongside personal and socio-cultural contexts. Methods (p.6) Study design This qualitative interview study will be informed by Strauss and Corbin's pragmatic approach to grounded theory (36, 37). This allows for (and recognises the likelihood of) researchers bringing prior knowledge to the process of refining theoretical concepts.
Introduction: References 17 and 18 provide data from over a decade ago. The authors may wish to look at more recent estimates. For example: https://doi.org/10.1080/14992027.2020.1773550	Thank you for the suggestion. We have now referenced the suggestion paper with more recent estimates. The following text has been added: Despite these benefits, hearing aid use is low and approximately 30% of people report wearing hearing aids some of the time and 20% do not wear them at all (17).
Aims and objectives: The text here differs from abstract, which suggests theory development. Here, the research aim is to develop a comprehensive contextualised explanation... Clarification required.	Please see our response above regarding the grounded theory approach we adopted and our revised aims and objectives.
Methods: Informed by grounded theory, but not grounded theory? Drawing on the principles of grounded theory. It is not clear which aspects of grounded theory methodology the authors will use and which they will omit. I would strongly suggest outlining their plans for the use of appropriate tools/guidance within this protocol to inform the reporting of ensuing study results, such as GUREGT for theoretical aspects, and e.g. COREQ as good practice for the reporting of qualitative research.	As highlighted above, our research draws on Strauss and Corbin's approach to grounded theory methodology and methods. Thank you for the suggestion of using a reporting tool. Please see attached a completed COREQ. Please note that given the prospective nature of the research study, not all questions were applicable.

Study research team – the inclusion of names is non-standard. What would be more useful within the protocol would be to outline the methods by which the influence of the study team will be monitored, reflected upon, and considered when undertaking the research and agreeing study results.	Thank you for this suggestion. We have removed the names of research teams members and replaced with author initials. We have integrated more information about the methods in which the study team’s influence will be monitored and considered in conducting and discussing/ interpreting findings (p.7-7): Research team and reflexivity For reflexivity, the study team comprises two female clinician-researchers (HP – an academic and hearing therapist with clinical experience in audiological rehabilitation and grounded theory methodology; SH – a clinical scientist, teaching fellow and researcher in audiology), three female researchers (GBOH – a teaching fellow and researcher with a background in sociology and interest in hearing research; RS – a Health Psychologist with experience of applied qualitative research; SS – a Health Psychologist with personal interest in hearing research) and a female PPI lead (JS) who is an expert in audiology by experience and writer. All the team either have first-hand experience of hearing loss or are closely related to someone who does. Participants will be interviewed by either HP, GBOC, SH or SS. At the start of each interview, the interviewers will undertake a ‘working alliance’ discussion agreeing participant and researcher roles, timing and nature of the interview. This will include the researcher’s interest and motivations in the topic. The study team will meet fortnightly to reflect on their role in shaping data collection and compare and contrast their interpretations of the data.

	Throughout the study, the wider HeLP research group and core PPI members will discuss interpretations of the findings for triangulation and work together to develop and link theoretical categories for the explanatory model.
I believe there are a number of omissions in the methods that the authors have likely considered and planned for. I use the COREQ checklist as reference to suggest that the authors should pre-specify within this protocol what plans they have in place for:  1. Reflexivity - when/how/impact (see comments on methods, above) 2. Methodological orientation and theory (see comments on methods, above) 3. How will you define and report participant non-participation? 4. Plans for data management 5. Plans for data analysis including software, methods, raters, validation ... 	Thank you for spotting these omissions. As suggested, we have completed a COREQ checklist attached.  1. Reflexivity – throughout the planned study, we will engage in reflexivity to examine how our own assumptions and beliefs shape the research process - please see responses above. 2. Methodological orientation and theory – as highlighted above, we will adopt Strauss and Corbin’s grounded theory approach will inform the methodological orientation and research methods, please see responses above. 3. Participant non-participation – our study will rely on voluntary participation and consent. We are not recording people who choose not to participate as they would be impossible to identify. 4. Plans for data management – personal data (contact details etc..) are held on a separate Aston University Box file and separate from research data. Besides clinicians in the Participant Identification Centres (PICs) sending the advertisements to home addresses, we will not access personal data. Transcribed interview research data and questionnaire data will be anonymised with identifiable places, clinical sites, names etc... removed from the transcripts (text addition on page 13). 5. Plans for data analysis –We will not use software to analyse data as we have four researchers in our team who will analyse their own subset of data and collate thematic findings. The axial and selective coding phases are conducted in person and individually providing opportunity for reflection, determination of negative cases and iterative interviews with negative cases. There is no validation in this case (this is qualitative work with a different epistemological stance). We aim for work to

	be credible in terms of explicit and well described data gathering & analysis procedures and fitting in terms of expanding on known phenomena with hearing loss experience. Our multiple researcher approach including our PPI members in analysis coding and interpretation allows for transparency in interpretation and analysis. Triangulation is a core part of our activity with multiple researcher and PPI members contributing a view to the interpretation which is justified by each researcher (text addition on page.10).
I am also unclear as to what the final output of the planned research will be (a developed theory or a contextual explanation - see comments on aims and objectives, above)? Further detail is required here	The final output of the planned research will be both - a novel theory, offering a theoretical understanding of the lived experience of hearing loss that has the potential to explain variations / differences between participants (cases). Please refer to our previous responses regarding our aims and grounded theory approach.
It is clear that this is part of a wider research programme. As such, describing the contribution of this research in relation to the wider research aims may be useful.	The contribution of the current research in relation to the wider aims of the research programme are now described in the introduction, page 6: This study forms part of a larger study – the Hearing Loss and Patient Reported Experience study (HeLP study) – the first study of its kind to prioritise patients’ lived experience of hearing loss to inform decision-making in audiology care planning and services. The findings from this qualitative work will inform the development of potential items for a novel Patient Reported Experience Measure (PREM) to be used in clinical practice to understand the work involved in living with hearing loss (‘illness work’) and hearing aids (‘treatment work’).
PPI: How has PPI shaped this study design/this protocol?	The following text has been added to reflect how our initial PPI work has shaped the study design and protocol (p.12) Our initial PPI work informed the development of the proposal in a number of ways. Firstly, our Aston group highlighted the need for researchers from the same culture and religious as participants to undertake PPI and research. As such, we will include a researcher with responsibility for PPI in

	marginalised groups and have recruited a researcher dedicated to South Asian groups. Secondly, our older group identified problems with hearing aids that inform the notion of ‘work’ (e.g. struggles to fit hearing aids, learning skills such as battery changing). These issues will be explored further by targeting PPI and recruitment in residential care settings where access to audiology services can be challenging. Thirdly, our older group noted the need to be flexible in where and how data gathering occurs and be mindful of travel and distance barriers for participants. This informed our plans to gather data in people’s homes, at clinics, university, and online/ telephone. Finally, our younger group advised we purchase amplifier headsets which could be sent out to participants for online/phone use, if required.
Twitter: Not sure these details are required?	We have removed Twitter details
Appendices: It is not a requirement to include the ethical approval letter as part of the published the study protocol.	We have removed the ethical approval letter
Reviewer: 2	
Can you please provide more detailed information about how your transcripts will be coded? I understand a subset of transcripts will be coded and compared by two researchers but how many people are initially coding the interviews?	We provide more detailed information (p.10) about the coding and analysis process: In line with Strauss and Corbin’s grounded theory procedures, data analysis and interviewing will be iterative, in which early analysis and findings will be compared to inform ongoing interviews. We will keep field notes of reflections and thoughts following interviews to support this. Whilst the notion of data ‘saturation’ has been widely critiqued as a neo-positivist indication of a singular ‘truth’; there is recognition of the importance of capturing a range of views, based on different contexts (39). The research team will review the data until we are confident that a wide range of experiences are captured and there is sufficient conceptual depth to generate meaningful insights to address the research aims (39). Meaning statements will be examined and every line of transcript will be coded by three coders (GBOC, SH, SS) to summarise key content (open coding). Codes will be compared across and between transcripts and will be collated into broader categories with properties and dimensions that capture the range of meanings

	(axial coding). Finally, the relationships between these categories will be explored to develop a conceptual model that explains the variation in experience (selective coding). Triangulation is a core part of our activity with multiple researcher and PPI members contributing a view to the interpretation which is justified by each researcher.
In the section Rigour in Analysis (pg 9-10) it reads like the theoretical connections will be formulated after the coding has been completed. Can you please clarify if this is what you mean or will you be developing the theory as the interview analysis progresses?	To clarify, we will be developing the theory as we interview participants and conduct analyses. Our findings will be used to target particular cases (e.g. negative cases and to challenge our developing conceptual model). Abstract (p.2): A grounded theory approach to concurrent data gathering and analysis will develop grouped codes and categories and link these to provide a novel theory to describe the experience of hearing loss. Text addition on page 10: In line with Strauss and Corbin's grounded theory procedures, data analysis and interviewing will be iterative, in which early analysis and findings will be compared to inform ongoing interviews. We will keep field notes of reflections and thoughts following interviews to support this. The HeLP qualitative research team (GBOC, HP, SH, JS, SS) will develop and refine the conceptual model whilst interviews and analysis progress to understand the effort required to manage hearing loss and highlight the variations in patient need at each life stage.
Please include information about theoretical saturation / data saturation, how do you know you have interviewed sufficient numbers?	In determining whether we have interviewed sufficient numbers, we will draw on guidance from Braun and Clarke (2021) about ceasing data collection/ recruitment activity. Thus, instead of focusing on reaching 'data saturation' to determine the sample size, we will regularly review the data (transcripts) until we are confident that interviews provide rich and comprehensive data across the target groups.

	The following text has been integrated to page 10: Whilst the notion of data ‘saturation’ has been widely critiqued as a neo-positivist indication of a singular ‘truth’; there is recognition of the importance of capturing a range of views, based on different contexts (39). The research team will review the data until we are confident that a wide range of experiences are captured and there is sufficient conceptual depth to generate meaningful insights to address the research aims (Braun and Clarke 2021). Page 8: Previous research experience informed our sample estimate of approximately 25-38 patients and 8-12 family members/carers (see Box 1) to capture a range of experiences and achieve sufficient information power, but this will be reviewed as data collection progresses.
Abstract: please specify acronym Patient and public involvement (PPI) when first used in abstract.	The acronym Patient and Public Involvement (PPI) is now specified in the abstract.
It wasn't clear why south Asian subgroups were singled out for recruitment, can you please provide rationale?	We targeted South Asian groups in both our PPI work and our participant recruitment strategy. Hearing loss among South Asian communities is particularly high, but uptake of audiology services is low. We were mindful of our local South Asian community living in Birmingham and wanted to involve them in the research. The following text has been added (p.11): Certain groups are more likely to be affected by hearing loss including people from South Asia (Indian, Pakistani, and Bangladeshi communities), older adults in residential care and adults with learning disabilities. Our PPI strategy reflects the diverse and heterogeneous nature of the population affected by hearing loss in the UK. We have directly

	targeted the following groups to expand our reach of PPI inclusion:  1. South Asian community groups – we have interest and agreement from these groups. We have also spoken to local religious centres, including but not limited to specific local Imams to enhance access to those from Muslim backgrounds. South Asian heritage people have both a disproportionately high risk of hearing loss and are underrepresented in patient populations in the UK (Dawes et al, 2014; WHO 2021)
Are you targeting only those who can speak English or will you be conducting interviews with interpreters?	We will be targeting both English speakers and non-English speakers. If required, we have access to interpreters. Our PPI work also informs us that in some circumstances family members may be preferable as interpreters to external services.
It is recommended that the semi-structured interview guide be provided as supplementary material.	The interview questions are now presented in Box 2.

VERSION 2 – REVIEW

REVIEWER	Henshaw, Helen University of Nottingham, School of Medicine
REVIEW RETURNED	20-Mar-2023

GENERAL COMMENTS	The aim in the abstract does not match the manuscript aim/objectives. It is unclear what and whom 'and to navigate audiology services' refers to. Please revise. The suggestion of including GUREGT/COREQ was in relation to the main study report: "I would strongly suggest outlining plans for the use of appropriate tools/guidance within this protocol to inform the reporting of ensuing study results, such as GUREGT for theoretical aspects, and e.g. COREQ as good practice for the reporting of qualitative research" (i.e. stating within the protocol that 'the publication of this research will be reported in accordance with GUREGT and COREQ...'). I'm not sure the supplemental file is necessary? Editor to advise. All other comments have been addressed. Thank you for the opportunity to review this interesting protocol!
--

REVIEWER	Laird, Emma C.
-----------------	----------------

	The University of Melbourne
REVIEW RETURNED	03-Apr-2023

GENERAL COMMENTS	Thank you for your revision of the manuscript, you have clarified your intended processes, particularly your application of grounded theory and reflexivity. I look forward to reading your results. I wanted to also draw your attention to some relevant publications related to your study should you wish to read / include. Knudsen, L. V., Nielsen, C., Kramer, S. E., Jones, L., & Laplante-Levesque, A. (2013, Mar). Client Labor: Adults with Hearing Impairment Describing Their Participation in Their Hearing Help-Seeking and Rehabilitation. J Am Acad Audiol, 24(3), 192-204. https://doi.org/10.3766/jaaa.24.3.5 Laird, E. C., Bennett, R. J., Barr, C. M., & Bryant, C. A. (2020). Experiences of Hearing Loss and Audiological Rehabilitation for Older Adults With Comorbid Psychological Symptoms: A Qualitative Study. American Journal of Audiology, 29(4), 809-824. https://doi.org/10.1044/2020_AJA-19-00123
--

VERSION 2 – AUTHOR RESPONSE

Reviewer 1	
Dr. Helen Henshaw, University of Nottingham, NIHR Nottingham Hearing Biomedical Research Unit	
The aim in the abstract does not match the manuscript aim/objectives. It is unclear what and whom 'and to navigate audiology services' refers to. Please revise.	We have revised the aim in the abstract (see below) to match with the aims/ objectives in the manuscript. Abstract: The aim of this qualitative study is to develop a novel theory to explain people's lived experience of hearing loss across the lifespan.
The suggestion of including GUREGT/COREQ was in relation to the main study report: "I would strongly suggest outlining plans for the use of appropriate tools/guidance within this protocol to inform the reporting of ensuing study results, such as GUREGT for theoretical aspects, and e.g. COREQ as good practice for the reporting of qualitative research" (i.e. stating within the protocol that 'the publication of this research will be reported in accordance with GUREGT and COREQ...).	Thank you for clarifying the suggestion regarding using GUREGT and COREQ for the main study results. We have added the following text on page 6: "This study will be conducted and reported in accordance with the consolidated criteria for reporting qualitative research checklist (COREQ)

	and the Guideline for Reporting and Evaluating Grounded Theory Research Studies (GUREGT).”
I'm not sure the supplemental file is necessary? Editor to advise.	Given the GUREGT and COREQ will be used for the main study results, we feel the supplemental file (COREQ checklist for the protocol) could be removed.
Reviewer 2: Dr. Emma C. Laird, The University of Melbourne	
I wanted to also draw your attention to some relevant publications related to your study should you wish to read / include. Knudsen, L. V., Nielsen, C., Kramer, S. E., Jones, L., & Laplante-Levesque, A. (2013, Mar). Client Labor: Adults with Hearing Impairment Describing Their Participation in Their Hearing Help-Seeking and Rehabilitation. J Am Acad Audiol, 24(3), 192-204. https://doi.org/10.3766/jaaa.24.3.5 Laird, E. C., Bennett, R. J., Barr, C. M., & Bryant, C. A. (2020). Experiences of Hearing Loss and Audiological Rehabilitation for Older Adults With Comorbid Psychological Symptoms: A Qualitative Study. American Journal of Audiology, 29(4), 809-824. https://doi.org/10.1044/2020_AJA-19-00123	Thank you for your positive comments about the revision. Thank you for drawing our attention to these relevant publications. We have now incorporated them into the introduction.